# Adherence to the Mediterranean Diet and Self-efficacy as Mediators in the Mediation of Sleep Quality and Grades in Nursing Students

**DOI:** 10.3390/nu12113265

**Published:** 2020-10-25

**Authors:** Isabel María Fernández-Medina, María Dolores Ruíz-Fernández, José Manuel Hernández-Padilla, José Granero-Molina, Cayetano Fernández-Sola, María del Mar Jiménez-Lasserrotte, María-Jesús Lirola, Alda Elena Cortés-Rodríguez, María Mar López-Rodríguez

**Affiliations:** 1Department of Nursing, Physiotherapy and Medicine, University of Almería, 04007 Almería, Spain; isabel_medina@ual.es (I.M.F.-M.); j.hernandez-padilla@ual.es (J.M.H.-P.); jgranero@ual.es (J.G.-M.); cfernan@ual.es (C.F.-S.); mjl095@ual.es (M.d.M.J.-L.); cra566@ual.es (A.E.C.-R.); mlr295@ual.es (M.M.L.-R.); 2Child and Midwifery Department, School of Health and Education, Middlesex University, London NW4 4BT, UK; 3Facultad de Ciencias de la Salud, Universidad Autónoma de Chile, 4780000 Temuco, Chile; 4Department of Education, University of Almeria, 04007 Almería, Spain; mariajesus.lirola@ual.es

**Keywords:** sleep, self-efficacy, Mediterranean diet, nursing students, academic performance, Pittsburgh sleep quality index, adherence to the Mediterranean diet, Baessler and Schwarzer general self-efficacy scale

## Abstract

University is a period in which students can experience a considerable amount of challenges that may influence their health lifestyles. The aim of this article is to discover the role of therapeutic adherence to the Mediterranean diet and self-efficacy as mediators in the relationship between sleep quality and the average grades of nursing students. The sample was made up of 334 nursing students, with a mean age of 21.84 years (SD = 6.24). Pittsburgh Sleep Quality questionnaires, adherence to the Mediterranean diet and the Baessler and Shwarzer General Self-efficacy Scale were administered. The results of the multiple mediational model determined that quality of sleep has a direct influence on academic performance. Adherence to the Mediterranean diet and quality of sleep have an effect on the degree of self-efficacy of nursing students. This study demonstrates that good sleep quality and adherence to the Mediterranean diet improve academic performance in nursing students. Future research should include multicenter longitudinal studies.

## 1. Introduction

Health-promoting behavior in terms of the actions that people carry out for the benefit of their health is considered the main strategy to improve an individual’s well-being and quality of life [1]. Essential components of health-promoting behavior are responsible for personal health, physical activity, nutrition, interpersonal relationships, self-fulfillment and stress management [2]. In developing countries, chronic non-communicable diseases, such as cancer, cardiovascular disease, diabetes and chronic obstructive pulmonary disease, are the leading cause of mortality and morbidity in adulthood [3]. According to the World Health Organization (WHO), most chronic non-communicable diseases are dependent on behavioral factors and therefore can be prevented [4].

Despite the fact that healthy types of behavior are acquired during the early years of life [5], adolescence and youth are critical periods for the incorporation of healthy habits that will continue throughout adulthood [6]. University is a period of life in which students experience a large number of challenges [7]. Adapting to living independently from their parents [8], developing new social networks [7] and achieving good marks can be stressful for university students [8]. Previous studies have shown the significant association between stress and poor dietary habits [9].

The Mediterranean diet, a food pattern traditional in Mediterranean countries, has been accepted as one of the healthiest dietary patterns [10]. The Mediterranean diet includes low consumption of saturated fat, salt and refined carbohydrates, with the use of olive oil as the main fat; a high consumption of fruit, vegetables, legumes, cereals and nuts; a moderate intake of dairy products and fish; and a low consumption of meat [10]. The combination of all these elements provides numerous health benefits, such as the prevention of cardiovascular disease, diabetes, breast and colorectal cancer and cognitive decline [11]. However, skipping breakfast, high levels of fast-food and sugary drink consumption and a low intake of fruit, vegetables and legumes seem to be common among university students [9,12]. Poor eating behavior has been significantly associated with inadequate sleep (˂7 h/night) [13].

The literature in the area has suggested that there is a relationship between academic achievement and healthy lifestyle behavior [14] so that frequent consumption of junk food, a sedentary life and insufficient sleep have been linked to poor academic marks [14,15]. On the other hand, self-efficacy, defined as the belief in one’s ability to perform a task [16], could have an influence on academic performance [17]. A high level of self-efficacy seems to decrease perceived stress and improve academic achievement [18].

Nurses have an important role in promoting community healthy lifestyles, and their own behavior might influence the health advice they give to patients [19]. Although nursing students recognize the importance of a healthy diet, quality sleep, physical activity and stress management to prevent diseases [20], their knowledge about healthy lifestyle behavior does not necessarily transfer to their own healthy lifestyle choices [21]. There are many barriers for nursing students to follow healthy lifestyles [21], and these include a lack of motivation and time, financial constraints and emotional stress due to challenging clinical situations [20].

Given that sleep and diet quality are both associated with the academic performance of students, the main aim of this study is to analyze the effect of therapeutic adherence to the Mediterranean diet and self-efficacy in the relationship between sleep quality and the average grades of nursing students. Therefore, the hypotheses of this study were: (1) good quality of sleep will improve adherence to the Mediterranean diet, which in turn will increase the self-efficacy of nursing students; (2) the quality of sleep will directly and positively influence the average marks of these students. The purpose of this study discovered the role of therapeutic adherence to the Mediterranean diet and self-efficacy as mediators in the relationship between sleep quality and the average grades of nursing students.

## 2. Materials and Methods

### 2.1. Design

This was a descriptive cross-sectional study.

### 2.2. Participants

The sample was made up of students following the nursing course at the University of Almería. Students who were enrolled in the current academic year at the time of the study were selected through the professors who taught that year. Students on university exchanges and those who did not want to take part in the study were excluded. Of a total of 480 students enrolled in the four courses of the undergraduate nursing degree, 334 students completed the questionnaires.

### 2.3. Instruments

The Pittsburgh Sleep Quality Index (PSQI) [22] was translated and validated for the Spanish population [23]. This instrument quantitatively assesses the quality of sleep in the previous month and consists of 19 items that combine to form 7 components related to sleep (subjective quality, latency, duration, habitual efficiency, disturbances, use of hypnotics and daytime dysfunction). The scores of the 7 components, each with a range of 0 to 3 points, are added up to form a global index of sleep quality. This global score ranges from 0 (no difficulties) to 21 points (difficulties). A score equal to or less than 5 corresponds to a good sleeper, and a score greater than 5 indicates a poor sleeper. The reliability of the questionnaire was 0.81 in patients and 0.67 in university students [23].

To assess adherence to the Mediterranean diet, the questionnaire designed by the Prevention with the Mediterranean Diet (PREDIMED) [24,25] research group was used. This questionnaire has been adapted to and validated for the Spanish population [26]. It consists of 14 items, with the score for each item being 0 if there is a negative connotation with respect to the Mediterranean diet or 1 if the connotation is positive. The higher the score, the greater the adherence to the Mediterranean diet [26].

The Baessler and Schwarzer General Self-efficacy Scale [27] has been adapted for the Spanish population by Sanjuán, Pérez and Bermúdez. It measures the degree of personal competence to effectively manage a series of stressful situations. The instrument consists of 10 items with responses on a 4-point Likert scale, which assess the feeling of personal competence to effectively handle a wide variety of stressful situations. The score ranges from 4 to 40 points. The higher the score, the higher the overall perceived self-efficacy. The internal consistency of the scale in the Spanish version is 0.87 in university students [27].

In addition, a series of sociodemographic and academic variables were collected: age, gender, academic year and average grades.

### 2.4. Procedure

The study was carried out from September to December 2019. The questionnaires described above were sent online to facilitate data collection. They were administered to the students on the virtual platform used by the students in the nursing degree in each of the four years of the course. Participation was completely voluntary and anonymous. The estimated completion time of the questionnaires was 15 min. The study obtained permission from the Ethics and Research Commission of the University of Almería (EFM-11/2019). The participants were informed about the aim of the study. Informed consent was requested prior to their participation. The confidentiality of the data and the anonymity of the participants respected the law in force at the national level (Organic Law 3/2018 on the protection of personal data and guarantee of digital rights). The principles of the Helsinki Declaration were respected.

### 2.5. Data Analysis

The qualitative variables were analyzed with measures of frequency and percentages, and measures of central tendency. The Pearson correlation test was performed in order to discover the relationship between the different variables (sleep quality index, adherence to the Mediterranean diet, self-efficacy and average academic grade). The *t* test for independent samples allowed us to know whether there were significant differences, at a significance level of 0.05. Subsequently, the effect of the mediation of therapeutic adherence to the Mediterranean diet and self-efficacy was analyzed in the relationship between sleep quality (independent variable) and average academic grade (dependent variable). A multiple mediational model was designed, with two mediating variables forming a causal chain. The mediation study was carried out with bootstrapping techniques. Ten thousand bootstrap samples were used at a 95% confidence level, yielding standardized effect measures and effect size. The macro PROCESS for SPSS was used for the mediation analysis according to Hayes28. Data analysis was performed with the statistical program SPSS version 25.0 (IBM Corp, Armonk, NY, USA).

## 3. Results

### Descriptive Data and Correlations Between Variables

The mean age of the participants was 21.84 (DT = 6.24) years. Of the participants, 79.6% (*n* = 266) were women and 20.4% (*n* = 68) were men. Further, 23.1% (*n* = 77) were in the first year, 31.4% (*n* = 105) in the second, 38% (*n* = 127) in the third and 7.2% (*n* = 24) in the fourth year. Table 1 shows the descriptions and correlations between the different variables. The correlations are significant and negative between the sleep quality index and adherence to the Mediterranean diet (r = −0.28; *p* ˂ 0.05), self-efficacy (r = −0.28; *p* ˂ 0.01) and the average score of the academic record (r = −0.18; *p* ˂ 0.01).

Table 2 shows the mean scores of the academic record according to the sleep quality index and adherence to the Mediterranean diet. Good sleepers have a significantly higher average score than poor sleepers. In terms of adherence to the Mediterranean diet, students with poor adherence to the Mediterranean diet are those who obtain significantly lower average scores, compared to those with good adherence.

Figure 1 shows the multiple mediational model with two moderating variables. This model responds to the following starting hypothesis: adherence to the Mediterranean diet (M1) and self-efficacy (M2) influence the relationship between the independent variable, quality of sleep (X), and the dependent variable, the mean grade of the academic record (Y). In this model, the quality of sleep (Y) has been used categorically (≤5 = good sleeper; ≥5 = poor sleeper).

Firstly, it is observed that the direct effect, including adherence to the Mediterranean diet (M1) and self-efficacy (M2), between sleep quality and mean academic score is significant (β = −0.31; *p* < 0.01) (Table 3). The total effect between both variables is also significant (β = −0.35; *p* < 0.001). Secondly, there is a significant relationship of sleep quality (X) on self-efficacy (M1) (β = −2.78; *p* ˂ 0.001). Finally, there is a positive relationship of adherence to the Mediterranean diet (M1) on self-efficacy (M2) (β = 0.41; *p* ˂ 0.05).

## 4. Discussion

This study explores the relationship between sleep quality and the average academic grade in nursing students, using therapeutic adherence to the Mediterranean diet and self-efficacy as mediators. Sleep plays a critical role in students’ physical and psychological health [28]. Following other studies, our results point out that quality of sleep is closely linked to academic performance in nursing students [29,30,31]. Poor sleep quality results in poor academic achievement [31]. Previous evidence has indicated that nursing students have a high prevalence of poor sleep quality, with insomnia being the main problem [32]. These sleep problems in nursing students could be attributed to factors such as prolonged Internet use [33], study at night, stress due to exams, coursework or clinical situations [34] and clinical shift work [35].

The impact of low sleep quality on cognitive performance and mental health has been previously documented [30]. A satisfactory sleep is essential to maintain motivation and attention and to memorize concepts [36,37]. Students who have a good night’s sleep have a greater capacity to learn and memorize than those who have sleep problems [36]. However, 10.5% of nursing students suffer daytime sleepiness because of insufficient sleep and an excessive workload [38]. The daytime sleepiness of students usually correlates with poor academic achievement and a lack of motivation [38], possibly related to the reduction of cognitive and attentional capacities, which determine reduced skill in managing the academic workload [39]. Furthermore, chronic sleep deprivation also has an influence on logical reasoning, problem solving and decision making [37], important skills for the clinical practice of nursing students.

We also observed that sleep disturbance negatively impacts self-efficacy. Consistent with our findings, James et al. [35] reported lower self-efficacy scores in nursing students with poor quality sleep, reducing their ability to safely carry out clinical practice. Self-efficacy is a useful predictor of motivation, and, for this reason, decreased self-efficacy scores include a reduction in motivation and difficulties in academic achievement [39].

In line with the results of recent investigations, we found a statistically significant correlation between poor sleep quality and reduced adherence to the Mediterranean diet [31]. Individuals with good sleep quality consume more carbohydrates and proteins and are less likely to eat high-calorie and fatty foods than individuals with poor sleep quality [40]. In addition, poor sleepers usually have irregular eating behavior [41]. Sleep deprivation brings changes in appetite-related hormones, increasing concentrations of ghrelin, a peptide that stimulates the appetite [42]. Furthermore, sleep problems alter brain connectivity and affect food decisions [42], which could explain the poor adherence to a Mediterranean diet. However, the direction of the relationship between sleep quality and dietary pattern is questionable. Evidence has demonstrated that poor sleepers have more appetite for high-calorie foods [41], and adherence to the Mediterranean diet improves sleep quality [31,43]. Two reasons could explain this. Firstly, evidence has underlined that some types of foods included in the Mediterranean diet, particularly fruit, vegetables, cereals and legumes, have positive effects on sleep induction and staying asleep [44]. These foods are rich in tryptophan, an essential amino acid and the precursor of melatonin, a protein involved in sleep efficiency and duration [44,45]. Secondly, adherence to the Mediterranean diet appears to exert a protective effect on stress [46], which is responsible for certain sleep problems.

In consonance with our results, adherence to the Mediterranean diet helps the academic performance of university students [47,48]. The study carried out by Esteban-Cornejo et al. [49] found that higher adherence to the Mediterranean diet correlated with a higher academic score. The nutrients provided by the Mediterranean diet could explain this association. The consumption of polyunsaturated fatty acids (included in fish and olives), vegetables and fruit, which are rich in flavonoids and Vitamins C and E, stimulates general cognition and the executive function [48,50]. The anti-inflammatory and antioxidant properties of the Mediterranean dietary pattern and its low glycemic index may contribute to better cognition [50]. In contrast, a diet full of saturated fat and simple sugar increases oxidate stress, which has been associated with difficulties in attention and concentration [51]. Previous studies have shown the implication of diet on mental health, especially on depression, that indirectly impacts on quality of life, self-efficacy and academic performance [52]. Considering that the Mediterranean diet affects cognitive performance, this statement could explain the beneficial effect that the Mediterranean diet has on self-efficacy, which was found in this study.

Another aspect to consider is the individual’s circadian preference, called a chronotype. Studies have shown that evening types have a reduced time and a low quality of sleep, daytime sleepiness and worse school performances [53]. Moreover, recent studies have shown that students with an evening chronotype had a lower adherence to the Mediterranean diet, reporting a significantly lower intake of fruits, vegetables, pulses, cereals and olive oil and higher breakfast skipping [54]. Therefore, this factor could be directly or indirectly affecting academic performance in line with this research.

Our results have several implications for nursing education. Poor sleep quality has implications not only for the health and the academic performance of nursing students but also for the safety of patients. It is therefore necessary for nursing students to acquire the appropriate knowledge about sleep hygiene and healthy habits and to understand the impact that their own healthy habits could have on their clinical practice. The university should develop programs or courses targeting healthy practices, mitigating sleep problems and increasing self-efficacy and the acquisition of strategies to improve academic achievement.

This study is not without limitations. One is that students from only one university participated, and thus our findings could not be generalized to include the entire population of nursing students. The lack of information on individual chronotype could have influenced the results. In addition, the data collection method might have skewed the nursing students’ responses. In future studies, it would be useful to determine the causality between the variables studied. Finally, multicenter longitudinal studies are needed to complete the results of this study. Although this study guides us on the association between the different variables, a longitudinal study is necessary to determine the cause-effect relationship. In addition, this research could be extended to other university centers.

## 5. Conclusions

The data confirms that good quality of sleep has a potentially positive effect on the academic performance of nursing students, as well as the mediating role that the Mediterranean diet plays between both variables. The healthy Mediterranean diet has also been shown to exert a positive influence on self-efficacy. The study results suggest the importance of improving healthy lifestyle habits in nursing students. Future longitudinal studies should be conducted in order to examine the effects of sleep patterns and the Mediterranean diet on the academic performance of nursing students.

## Figures and Tables

**Figure 1 nutrients-12-03265-f001:**
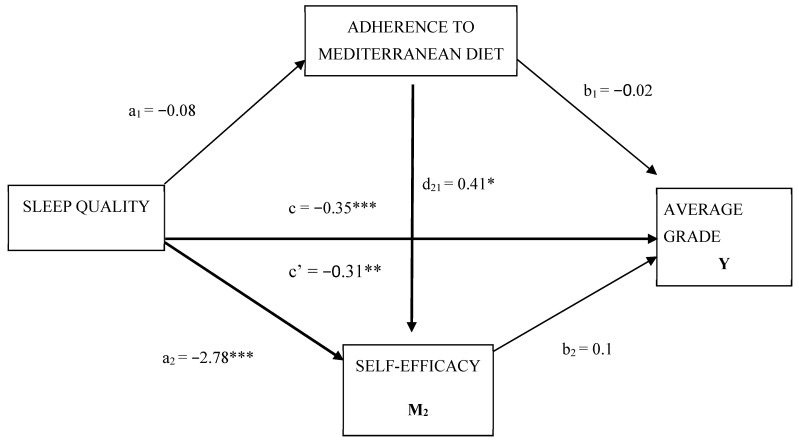
MMultiple model of mediation of adherence to the Mediterranean diet and self-efficacy in the relationship between the quality of sleep and the average grades of the students. X = independent variable; M1 = moderating variable 1; M2 = moderating variable 2; Y = dependent variable; a1= effect sleep quality on adherence to Mediterranean diet; b1= effect adherence to Mediterranean diet on average grade; a1 = effect sleep quality on self-efficacy; b2= effect self-efficacy on average grade; d21= effect adherence to Mediterranean diet on self-efficacy; c’ = direct effect of X on Y with M1 and M2; c = total effect of X on Y; * *p* < 0.05; ** *p* < 0.01; *** *p* < 0.001.

**Table 1 nutrients-12-03265-t001:** Descriptive and bivariate correlations between the different variables under study.

Variables	N	M (DT)	1	2	3	4
1. Sleep quality index	284	6.66 (3.59)	-	−0.13 *	−0.28 **	−0.18 **
2. Adherence to the Mediterranean diet	334	6.35 (1.74)		-	0.12 *	−0.06
3. Self-efficacy	334	29.41 (5.04)			-	0.08
4. Average score of the academic grade	334	7.12 (0.91)				-

** *p* < 0.01; * *p* < 0.05.

**Table 2 nutrients-12-03265-t002:** Descriptive average academic grade, according to sleep quality and adherence to the Mediterranean diet.

Variables		N	M	DT	t	*p*	d
Sleep Quality Index	Good sleeper	128	7.37	0.85	3.42	0.001 **	0.35
	Poor sleeper	152	7.02	0.88			
Adherence to Mediterranean Diet.	Good adherence	35	6.64	1.32	3.32	0.001 **	0.53
	Poor adherence	299	7.18	0.84			

** *p* < 0.001.

**Table 3 nutrients-12-03265-t003:** Total, direct and indirect effect: multiple model of mediation.

Effects	B	SE	t	*p*	CI (95%)
Total effect X Y	−0.357	0.104	−3.426	0.000	(−0.562; −0.152)
Direct effect X Y	−0.318	0.108	−2.937	0.003	(−0.532; −0.152)
Ind 1 X M_1_ Y	0.00	0.00			(−0.015; 0.025)
Ind 2 X M_2_ Y	−0.03	0.02			(−0.111; 0.011)
Ind 3 X M_1_ M_2_ Y	−0.00	0.00			(−0.004; 0.002)

X (Sleep Quality) = dependent variable; Y (average academic grade) = independent variable; M1 (adherence to the Mediterranean diet) = moderating variable 1; M2 (self-efficacy) = moderator variable 2; Ind = indirect effect; B = non-standardized regression coefficient; SE = standard error; *t* = student’s t; CI: confidence interval.

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
