# Peer review of "Adherence to the Mediterranean Diet and Self-efficacy as Mediators in the Mediation of Sleep Quality and Grades in Nursing Students"

_nutrients, 2020, doi:10.3390/nu12113265_

Round 1

Reviewer 1 Report

The authors stated that the study received ethical approval. Please add the referral ID number. 

How did the students are enrolled in the study? 

Were the average grades self-reported? if yes, please add.

In the discussion Authors stated (lines 218-220) "In contrast, a diet full of saturated fat and simple sugar increases oxidate stress, which has been associated with difficulties in attention and concentration". Considering this statement, Authors could take into account the implication of diet on mental health and particularly on depression, that indereclty impact on quality of life, self-efficacy and academic performance. I suggest authors to refer to  doi: 10.3390/ijerph17186686.

Author Response

1. The authors stated that the study received ethical approval. Please add the referral ID number.

Response: Thank you for your comments. We have added the referral ID number of ethical approval.

2. How did the students are enrolled in the study?

Response: Thank you for your comments. We have added the next information in participants section: Students who were enrolled in the current academic year at the time of the study were selected, through the professors who taught that year.

3. Were the average grades self-reported? if yes, please add.

Response: Thank for your consideration. The average grades weren’t self-reported by students.

4. In the discussion Authors stated (lines 218-220) "In contrast, a diet full of saturated fat and simple sugar increases oxidate stress, which has been associated with difficulties in attention and concentration". Considering this statement, Authors could take into account the implication of diet on mental health and particularly on depression, that indereclty impact on quality of life, self-efficacy and academic performance. I suggest authors to refer to doi: 10.3390/ijerph17186686.

Response: Thank you for your consideration. We have followed your instructions and we have referenced the study in the text. We have added:

Previous studies have shown the implication of diet on mental health, especially on depression, that indirectly impact on quality of life, self-efficacy and academic performance (52).

Reviewer 2 Report

1) Please provide a Title which refers to the content of the results/ conclusions.

2) Key Words: Please add the Scales that have been used in this Study.

3) Introduction: (Line 42 ) Please provide more details regarding MedDiet.

                           (Line 43) why is  the healthiest dietary pattern ?

Please support your statement with more studies, regarding adherence to MedDiet as beneficial for mental health status, stress, mood and overall health.

4) 2.2 Participants: Please remove lines 77 to 79 (The Mean age .. in the fourth year) and add to 3. Results

Author Response

1) Please provide a Title which refers to the content of the results/ conclusions.

Response: Thank you for your comments. The title has been modified according to the results of the study.  

2) Key Words: Please add the Scales that have been used in this Study.

Response: Thank you for your considerations. We have added that scales that have been used in this study as keywords.

3) Introduction: (Line 42) Please provide more details regarding MedDiet.

                           (Line 43) why is the healthiest dietary pattern ?

Response: Thank you for your considerations. Please support your statement with more studies, regarding adherence to MedDiet as beneficial for mental health status, stress, mood and overall health.

4) 2.2 Participants: Please remove lines 77 to 79 (The Mean age .in the fourth year) and add to 3. Results

Response:  Thank you for your comments. We have removed indicated lines.

Reviewer 3 Report

The article "Adherence to the Mediterranean diet and self-efficacy as mediators in the relationship between sleep quality and average grades in nursing students." talks about adherence to the Mediterranean diet and self-efficacy in the mediation of sleep quality and grades in nursing students.
The study has a number of methodological shortcomings that are detailed below:

1. The citations in the text are not correct as indicated by the magazine.

2. In the abstract, the authors point out that nursing students know the importance of healthy habits. This statement is very general and is not correct. The study subjects are from 1st to 4th grade in nursing with the limitations not indicated in the study that this entails. The collection period was from September to December, so the first-year students (77) still did not have this knowledge since they had just started their career.
On the other hand, the sample is only collected from nursing, there are more careers in Health Sciences that may have this knowledge and it would be interesting to study.

3. In lines 94-95 there are missing quotes to support what is mentioned.

4. Lines 108-109 are missing the ethics committee acceptance code.

5. In the discussion there are missing articles in Spanish university students on the Mediterranean diet. There are several articles dealing with this topic in Spain. One of them is the article "Associations between Chronotype, Adherence to the Mediterranean Diet and Sexual Opinion among University Students".

6. In the introduction it would be convenient to close it at the end for the purpose of the study.

Author Response

  1. The citations in the text are not correct as indicated by the magazine.

Response: Thank you for your considerations. The citations have been corrected according to the magazine.

2. In the abstract, the authors point out that nursing students know the importance of healthy habits. This statement is very general and is not correct. The study subjects are from 1st to 4th grade in nursing with the limitations not indicated in the study that this entails. The collection period was from September to December, so the first-year students (77) still did not have this knowledge since they had just started their career.
On the other hand, the sample is only collected from nursing, there are more careers in Health Sciences that may have this knowledge and it would be interesting to study.

Response: Thank you for your considerations. I agree with you. The statement about nursing students in the abstract is not correct. We have replaced it for other statement more appropriate about challenges in university.

3. In lines 94-95 there are missing quotes to support what is mentioned.

Response: Thank you for your considerations. Citations have been included in lines 94-95.

4. Lines 108-109 are missing the ethics committee acceptance code.

Response: Thank you for your considerations.I have included the ethics committee acceptance code.

5. In the discussion there are missing articles in Spanish university students on the Mediterranean diet. There are several articles dealing with this topic in Spain. One of them is the article "Associations between Chronotype, Adherence to the Mediterranean Diet and Sexual Opinion among University Students".

Response: Thank you for your considerations. We have included more articles in Spanish university students on the Mediterranean diet in the discussion.

6. In the introduction it would be convenient to close it at the end for the purpose of the study.

Response: Thank you for your considerations.The introduction has been closed it at the end for the purpose of the study.

Reviewer 4 Report

The authors intended to verify the hypothesis that: 1) good quality of sleep will improve adherence to the Mediterranean diet, which in turn will increase the self-efficacy of nursing students; 2) the quality of sleep will directly and positively influence the average marks of the students. The study is interesting, and well conducted. However, no mention is given (introduction and/or discussion) to individual circadian preference (chronotype). It is widely known that evening types (owls) have a reduced time of sleep and a low quality, daytime sleepiness, and worst school performances (Fabbian et al, Chronobiol Int 2016). Moreover, recent studies showed that students with an evening chronotype had a lower adherence to the Mediterranean diet, reporting a significantly lower intake of fruits, vegetables, pulses, cereals, olive oil, and higher breakfast skipping (Rodriguez-Munoz et al, Nutrients 2020). I would suggest to add these references and discuss this topic. The lack of information on individual chronotype could be possibly included in the limitations section.

Author Response

The authors intended to verify the hypothesis that: 1) good quality of sleep will improve adherence to the Mediterranean diet, which in turn will increase the self-efficacy of nursing students; 2) the quality of sleep will directly and positively influence the average marks of the students. The study is interesting, and well conducted. However, no mention is given (introduction and/or discussion) to individual circadian preference (chronotype). It is widely known that evening types (owls) have a reduced time of sleep and a low quality, daytime sleepiness, and worst school performances (Fabbian et al, Chronobiol Int 2016). Moreover, recent studies showed that students with an evening chronotype had a lower adherence to the Mediterranean diet, reporting a significantly lower intake of fruits, vegetables, pulses, cereals, olive oil, and higher breakfast skipping (Rodriguez-Munoz et al, Nutrients 2020). I would suggest to add these references and discuss this topic. The lack of information on individual chronotype could be possibly included in the limitations section.

Response: Thank you for your contribution. We have this topic in discussion part and we have added the references.We have included in the limitations section that the lack of information on individual chronotype could have influenced the results.

Please see the atachment.
